# Structure-Functional Characteristics of the Svx Protein—The Virulence Factor of the Phytopathogenic Bacterium *Pectobacterium atrosepticum*

**DOI:** 10.3390/ijms23136914

**Published:** 2022-06-21

**Authors:** Natalia Tendiuk, Tatiana Konnova, Olga Petrova, Elena Osipova, Timur Mukhametzyanov, Olga Makshakova, Vladimir Gorshkov

**Affiliations:** 1Kazan Institute of Biochemistry and Biophysics, FRC Kazan Scientific Center of RAS, 420111 Kazan, Russia; natasha.tendjuk@rambler.ru (N.T.); tatiana.a.konnova@gmail.com (T.K.); poe60@mail.ru (O.P.); eva-0@mail.ru (E.O.); olga.makshakova@kibb.knc.ru (O.M.); 2Kazan Federal University, 420008 Kazan, Russia; timur.mukhametzyanov@kpfu.ru

**Keywords:** gluzincin extracellular metallopeptidases, molecular docking, *Pectobacterium*, plant-pathogen interactions, protein structure, Svx proteins, virulence factors

## Abstract

The Svx proteins are virulence factors of phytopathogenic bacteria of the *Pectobacterium* genus. The specific functions of these proteins are unknown. Here we show that most of the phytopathogenic species of *Pectobacterium*, *Dickeya*, and *Xanthomonas* genera have genes encoding Svx proteins, as well as some plant-non-associated species of different bacterial genera. As such, the Svx-like proteins of phytopathogenic species form a distinct clade, pointing to the directed evolution of these proteins to provide effective interactions with plants. To get a better insight into the structure and functions of the Svx proteins, we analyzed the Svx of *Pectobacterium atrosepticum* (*Pba*)—an extracellular virulence factor secreted into the host plant cell wall (PCW). Using in silico analyses and by obtaining and analyzing the recombinant *Pba* Svx and its mutant forms, we showed that this protein was a gluzincin metallopeptidase. The 3D structure model of the *Pba* Svx was built and benchmarked against the experimental overall secondary structure content. Structure-based substrate specificity analysis using molecular docking revealed that the *Pba* Svx substrate-binding pocket might accept α-glycosylated proteins represented in the PCW by extensins—proteins that strengthen the PCW. Thus, these results elucidate the way in which the *Pba* Svx may contribute to the *Pba* virulence.

## 1. Introduction

Members of the soft rot *Pectobacteriaceae* (SRP), including *Pectobacterium atrosepticum* (*Pba*), cause severe plant diseases [1]. These bacteria produce multiple plant cell wall degrading enzymes (PCWDEs), leading to host plant tissue maceration [2]. However, PCWDEs alone are not sufficient to provide full virulence to *Pectobacterium* species, and additional virulence factors significantly contribute to the manifestation of disease caused by these bacteria, e.g., Nip protein [3,4], type 3 secretion system (T3SS) [5,6], type 6 secretion system (T6SS) [7,8], coronafacic acid [9,10,11], and Svx protein [12].

The Svx protein is one of the “black horses” of plant-pathogen interactions. The homolog of this protein (AvrXca) has been first described as an avirulence factor of *Xanthomonas campestis* pv. *raphani* 1067 [13]. The transfer of a gene *avrXca* of the avirulent strain *X. campestis* pv. *raphani* 1067 to virulent strains of *X. campestis* resulted in the reduced virulence of the latter. However, the *avrXca* genes were also identified in virulent *Xanthomonas* strains [13], indicating that different AvrXca proteins may act in different ways in terms of plant-pathogen interactions.

In *Pba*, the Svx protein has been shown to act as a virulence factor since the knockout of the *svx* gene reduced the ability of the pathogen to cause disease [12]. The known Svx homologs (Svx of *Pectobacterium*, AvrXca of *Xanthomonas*, and AvrL of another SRP genus *Dickeya*) are secreted proteins [12,13,14]. In *Pba* and *Dickeya dadantii*, these proteins are transferred via the type two secretion system (T2SS), as well as most of the PCWDEs, to the environment, including the host plant apoplast [12,14]. Similar to many virulence factors, the production of the *Pba* Svx protein is induced by quorum sensing and host plant metabolites, including pectic compounds [7,12]. The expression of the *Pba svx* gene is highly upregulated *in planta* at both asymptomatic and symptomatic colonization stages [10].

Although information about the regulation of the Svx protein production and its secretion exists, nothing is known about the mechanism of action of this virulence factor. Therefore, the aim of our study was to get a better insight into the mechanism of action of the *Pba* Svx protein during plant-microbe interactions. We analyzed the diversity of the Svx-like proteins and performed their phylogenetic analysis. By using different in silico approaches, we predicted functional domains within the *Pba* Svx protein, one of which, the peptidase domain, was verified using the recombinant protein and its mutant forms. Furthermore, the 3D structure model of the *Pba* Svx protein was built and benchmarked against the experimental overall secondary structure content of the protein. The structure-based substrate specificity analysis using molecular docking distinguished a potential target of the *Pba* Svx protein among the host plant proteins.

## 2. Results

### 2.1. Phylogenetic Analysis of the Svx-like Proteins

A phylogenetic analysis was performed to characterize the diversity of the Svx-like proteins and assess possible relationships between the features of their primary structure and the ecology of the bacteria producing them. Using the DELTA-BLAST algorithm [15,16], we identified 69 Svx-like proteins that had a coverage level higher than 76% and an identity of 40.81% to 97.64% (E-value < 7–128) with the *Pba* Svx protein (WP_011092533.1). These proteins were found in Gammaproteobacteria (genera *Pectobacterium*, *Dickeya*, *Xanthomonas*, *Zymobacter*, *Samsonia*, *Vibrio*, *Teredinibacter*, *Microbulbifer*), Alphaproteobacteria (genera *Asticcacaulis*, *Sphingomonas*, *Novosphingobium*, *Sphingobium*), and Deltaproteobacteria (genera *Corallococcus*, *Cystobacter*, *Stigmatella*, *Pyxidicoccus*) (Figure 1).

The Svx proteins of phytopathogenic species of Gammaproteobacteria (*Pectobacterium* species, *Dickeya* species, *Xanthomonas* species, and *Samsonia erythrinae*), along with a similar protein of a plant endophytic Gammaproteobacterium *Zymobacter palmae* [1,17,18,19,20], formed a separate clade G (Figure 1). Svx-like proteins were found in all species of the *Pectobacterium* and *Dickeya* (except *D. poaceiphila*) genera. The genomes of some *Dickeya* species contain two slightly distinct genes encoding the Svx-like proteins. Within the *Xanthomonas* genera, in most (*X. campestris*, *X. transculens*, *X. arboricola*, *X. dyei*, *X. hortorum*, *X. cucurbitae*, *X. floridensis*, *X. sacchari*, *X. sontii*, *X. citri*, *X. codiaei*, *X. nasturtii*, *X. melonis*, *X. pisi*, *X. vesicatoria*, *X. cannabis*), but not all species, genes encoding Svx-like proteins were revealed.

The Svx-like proteins of free-living Alphaproteobacteria formed two other clades (A-1 and A-2) (Figure 1). The fourth clade (DG) was represented by the proteins of Deltaproteobacteria and non-phytopathogenic species of Gammaproteobacteria. Thus, the Svx-like proteins of Gammaproteobacteria split into phylogenetically distant groups (G and DG): the proteins of phytopathogenic and one plant endophytic Gammaproteobacteria constituted the first clade (G), while those of non-phytopathogenic/non-plant endophytic Gammaproteobacteria—*Vibrio quintilis* (inhabitant of sea water, [21]), *Vibrio mangrovi* (inhabitant of the rhizosphere of mangrove-associated wild rice, [22]), *Microbulbifer rhizospaerae* (inhabitant of the rhizosphere of halophilic plants, [23]), and *Teredinibacter turnerae* (symbiont of mollusk, [24])—located in the fourth clade (DG). Based on multi-protein phylogeny [25] and 16S rRNA sequence, *Pectobacterium*, *Dickeya*, and *Samsonia* are phylogenetically closer to *Vibrio*, *Teredinibacter*, and *Microbulbifer* than to *Xanthomonas* species. Despite that, the Svx proteins of *Pectobacterium*, *Dickeya*, and *Samsonia* were phylogenetically closer to the Svx proteins of *Xanthomonas* species than to those of the *Vibrio*, *Teredinibacter*, and *Microbulbifer* species.

The differences between the Svx-like proteins of the G and DG clades were related to the presence of insertions and deletions in the representatives of the DG clade. In some of the proteins of the DG clade, additional functional domains were revealed. In addition to the metallopeptidase domain and left-handed parallel beta-helix (LbH) domain (acyltransferase-like domain) revealed in the Svx-like proteins of all clades, in the Svx-like proteins of *Vibrio mangrove*, two fibronectin type III-like domains were found, while in the Svx-like proteins of *Teredinibacter turnerae*, a cellulose-binding II/carbohydrate binding (CBM 35) domain was present (Appendix A).

In almost all of the revealed Svx-like proteins (except for *Zymobacter palmae*), signal peptides were found, indicating that these proteins were either extracellular or plasma membrane-localized. The Svx-like proteins of *Pectobacterium*, *Dickeya*, *Samsonia*, *Xanthomonas*, *Vibrio*, *Teredinibacter*, *Novosphingobium*, and *Stigmatella* have Sec/SPI type signal peptides, while the Svx-like proteins of *Sphingobium* possess Tat/SPI type signal peptides. Both Sec/SPI and Tat/SPI signal peptides are usually characteristic of extracellular proteins. In the Svx-like proteins of *Sphingomonas*, *Asticcacaulis*, *Cystobacter*, *Corallococcus*, *Pyxidicoccus*, and *Microbulbifer*, the Sec/SPII signal peptides characteristic of membrane lipoproteins [27] were revealed.

### 2.2. Identification and Experimental Verification of the Peptidase Domain of the Svx Protein of P. atrosepticum

The analysis of the primary structure of the *Pba* Svx protein with I-Tasser [28,29] and Phyre 2 [30,31] servers revealed the peptidase domain in the region of 43–445 amino acids. The revealed peptidase domain was similar to the peptidase domains of gluzincin extracellular metallopeptidases, whose catalytic sites are formed by the conservative zinc-binding motif HEXXHX(8,28)E (X–denotes any amino acid) [32]. The alignment of the amino acid sequence of the *Pba* Svx protein with other bacterial gluzincin metallopeptidases showed that the *Pba* Svx protein possessed the HEXXHX(8,28)E motif (Figure 2), supporting the predicted peptidase activity of the *Pba* Svx protein.

In order to verify whether the *Pba* Svx protein is a metallopeptidase, the recombinant protein was obtained. The enzymatic assay with azocasein as a substrate proved that the *Pba* Svx protein possessed peptidase activity, which was most pronounced at a pH of 7.5 and 40 °C (Figure 3).

So as to confirm that zinc-binding motif HEXXHX(8,28)E determined the peptidase activity of the *Pba* Svx protein, the influence of Zn^2+^ ions (ZnSO_4_) and metal chelator ethylenediaminetetraacetic acid (EDTA) on the peptidase activity of the *Pba* Svx protein was determined. Additionally, we obtained two mutant Svx proteins: SvxΔE141A and SvxΔE167A, in which glutamic acid residues of the HEXXHX(8,28)E motif were substituted for alanine, and the activity of mutant enzymes was compared to that of the wild type (Figure 4).

The addition of 1 mM ZnSO_4_ to the *Pba* Svx protein preparation increased its peptidase activity by 20%, while the presence of 1 mM EDTA totally inhibited the peptidase activity (Figure 4). The addition of 1 mM or 2 mM ZnSO_4_ gradually restored the peptidase activity of *Pba* Svx protein in the presence of 1 mM EDTA. The mutant proteins SvxΔE141A and SvxΔE167A displayed 3.1- and 3.5-fold lower peptidase activity, respectively, compared to that of the wild-type protein (Figure 4). Thus, our results showed that the *Pba* Svx protein was a gluzincin metallopeptidase, whose active site was formed by the conservative zinc-binding motif HEXXHX(22)E.

### 2.3. Study of the Secondary Structure of the Svx Protein of P. atrosepticum

To characterize the secondary structure of the *Pba* Svx, in silico analyses of its amino acid sequence were performed using the PSIPRED [43,44] and AlphaFold 2 [45,46] servers, and its solution conformation was studied using far-UV circular dichroism (CD) spectroscopy. The in silico analyses revealed alpha-helices, beta-sheets, turns, and unordered structures within the *Pba* Svx protein (Table 1). The CD spectra of the *Pba* Svx protein displayed a positive peak around 195 nm and a negative peak around 215 nm, both of which are characteristic of the beta-sheet structural motifs (Figure 5). A broad negative ellipticity between 208 and 222 nm and small distinct minima at 210 and 222 nm indicated that the *Pba* Svx protein possessed some alpha-helical structures under the experimental conditions. No concentration-dependent changes in the CD spectra of the *Pba* Svx protein were observed within the studied range (0.1–0.4 mg/mL) (Figure 5).

In order to determine the content of the secondary structures in the *Pba* Svx protein, the Dichroweb internet server was used. The CONTINLL method [47,48] was chosen to estimate the content of the secondary structures in the *Pba* Svx protein since it gave the best fitting of the estimated curve to the experimental points with a normalized root mean square deviation (NRMSD) ≤ 0.1. Under the studied conditions, beta-sheets were the main type (32.3 ± 0.98%) of regular conformations of the *Pba* Svx protein (Table 1). The alpha-helices contributed less to the secondary structure of the *Pba* Svx protein (17.0 ± 0.49). Turns and unordered structures constituted 21.2 ± 0.28 and 29.5 ± 0.86%, respectively, of the *Pba* Svx protein (Table 1). Thus, the results of the in silico and CD-spectroscopy analyses were in good agreement, indicating that the Svx protein was predominantly composed of beta-sheet structures with some alpha-helices (Table 1).

### 2.4. Prediction of Possible Substrates of the Svx Protein of P. atrosepticum

So as to propose possible substrates of the *Pba* Svx protein, its spatial structure was built using the AlphaFold 2 service (Appendix A) [45,46]. The model contained two structural domains: a peptidase domain formed by α-helices and β-strands and a bacterial transferase hexapeptide (acyltransferase-like, PF00132) domain formed by a left-handed parallel β-helix (Figure 6).

The quality of the Alpha Fold 2 structure is high, with the predicted local-distance difference test (pLDDT) above 70% for the majority of the residues (Appendix A). For further analysis, we used only the metallopeptidase domain of this structure. After structure minimization, the quality of the obtained model was checked using the PROCHECK server [49,50]. The G-scores were as follows: −0.29 for the overall score, −0.25 for dihedrals, and −0.37 for covalent bonds, which reveals the high quality of the model. Eighty percent of the backbone phi-psi angles were located in the most favored regions of the Ramachandran plot, around 19% of the angles occupied allowed regions, while less than 1% of the angles were located at the border of allowed and disallowed regions; the residues located at the border were found in flexible loops. The content of the secondary structures in the computer model was in good agreement with that estimated using CD-spectroscopy and predicted by the PSIPRED algorithm (Table 1).

The search for the structural homologs of the *Pba* Svx protein over gluzincin family metallopeptidases showed that the peptidase domain of the *Pba* Svx protein had the closest folding to the peptidase domain of the O-glycopeptidase ZmpB of *Clostridium perfringens* (Appendix A). ZmpB is a member of the M60 family (PF13402) that includes viral enhancins and enhancin-like peptidases from animal host mucosa-associated prokaryotic and eukaryotic microbes [51]. The active site of M60 peptidases contains a catalytic site and a substrate binding site. The former includes the zinc-binding motif HEXXHX(8,28)E, which is located on two alpha-helices. The latter is a carbohydrate binding site located next to the catalytic site and formed by amino acid residues that take part in recognition of N-acetyl-galactosamine in glycoprotein substrates (such as mucin) [39,52].

The *Pba* Svx active site contains amino acid residues similar to those found in ZmpB glycopeptidase in both the zinc-binding motif (HEXXHX(8,28)E) and the carbohydrate-binding site (Figure 7). In addition, the amino acid residues W136, G137, and N171 of the carbohydrate-binding site are highly conserved (>80%) among Svx-homologs, implying that they also bind substrates in the majority of the Svx proteins, including the *Pba* Svx. Therefore, from a structural point of view, it is reasonable to presume that *Pba* Svx is a glycopeptidase hydrolyzing the glycosylated proteins. Moreover, since this protein is secreted by *Pba* into the host plant apoplast, the substrates of the *Pba* Svx are likely to be the glycosylated proteins of the plant cell wall.

Plant cell wall (PCW) proteins can be O-glycosylated or N-glycosylated [53]. O-glycosylated PCW proteins can be split into two major groups according to their glycosylation pattern. The first group includes extensins, hydroxyproline-rich glycoproteins, and their chimeras, the glycosylation motif of which is formed by α-galactose linked to serine followed by four hydroxyprolines (Ser-Hyp-Hyp-Hyp-Hyp) glycosylated with β-arabinose chains [53]. The second group contains arabinogalactans, in which the first carbohydrate residue of the glycosylation motif—the galactose—is attached to the hydroxyproline residue by the β-linkage [54]. In various N-glycosylated PCW proteins, the N-acetyl-glucosamine of the glycosylation motif is attached to the asparagine residue by the β-linkage [54].

The glycopeptidases from the M60 family, including ZmpB, interact with mammal O-glycosylated proteins (e.g., mucin), in which a conserved O-glycosylation motif—N-acetyl-galactosamine residue α-linked with serine/threonine residue—is recognized by the enzymes [39]. Since galactose is structurally related to N-acetyl-galactosamine, we proposed that the *Pba* Svx protein can bind to serine amino acid O-glycosylated with α-galactose and break down the peptide bond between serine and hydroxyproline residues in extensins or extensin-like domain-containing proteins. In order to determine whether the *Pba* Svx can indeed accommodate α-galactosylated peptides, the molecular docking of the *Pba* Svx protein with α-Gal-1-SAPGG was performed. The α-Gal-1-SAPGG ligand was built based on the structure of the ligand previously used for the study of ZmpB-substrate interaction [39].

In the X-ray structure of the ZmpB-O-glycozylated peptide complex, the N-atom of the glycosylated amino acid was the closest one to Zn^2+^, located at a distance of 3.8 Å from the ion [39]. In the *Pba* Svx, the binding pocket for α-galacatose was formed by the W136, H140, N171, and L113 residues; the N-atom of the serine residue was located at 2.1 from Zn^2+^ (Figure 8). The calculated dissociation constant of the *Pba* Svx-α-Gal-1-SAPGG complex was around 90 mM. The interactions between the *Pba* Svx substrate binding site and the ligand α-Gal-1-SAPGG were mainly determined by the recognition of the carbohydrate part of the substrate (Figure 8). The *Pba* Svx did not form contacts with the pentapeptide portion of the α-Gal-1-SAPGG ligand (similar to ZmpB, [39]), suggesting that the *Pba* Svx protein did not display specificity to the peptide moiety (Figure 8).

To check whether the *Pba* Svx protein had the potential to hydrolyze not only α-glycosylated proteins but also two other groups of PCW glycosylated proteins (arabinogalactans β-glycosylated with galactose and proteins β-glycosylated with N-acetyl-glucosamine), we performed an analysis of the interactions between the *Pba* Svx protein and β-glycosylated amino acid residues. Since the *Pba* Svx protein had an affinity for the α-galactosyl moiety of glycosylated peptides/proteins rather than for their peptide moiety (Figure 8 and Figure 9A), we assumed that the binding of other glycosylated species by the *Pba* Svx protein would retain the orientation of the sugar moiety in the carbohydrate-binding site. To have a preliminary structure-based vision of the interactions of the *Pba* Svx with β-galactosylated hydroxyproline (characteristic of arabinogalactans) or asparagine β-linked to N-acetyl-glucosamine, the sugar parts of these ligands were superimposed with the α-galactosyl moiety of α-Gal-1-SAPGG. The O- and N-atoms in the β-anomeric positions of β-galactose and β-N-acetyl-galactosamine overlapped with the *Pba* Svx protein, respectively (Figure 9B,C). The aglycon linked to the β-anomers overlapped with the *Pba* Svx protein, forming large clashes; with this, the formation of clashes was predominantly due to the H140 position, which is important for the Zn^2+^ binding. Thus, the topology of the *Pba* Svx protein’s carbohydrate binding site allows it to bind α-glycosylated amino acid residues; however, the affinity for β-glycosylated amino acid residues is expected to be low, if present. Therefore, we conclude that the *Pba* Svx protein can possibly recognize α-galactosylated amino acid residues of extensins and hydrolyze these structural PCW glycoproteins.

## 3. Discussion

Our study provides the first data on the functions of a representative of a large but uncharacterized protein family—the Svx-like proteins. The genes encoding Svx-like proteins are widely distributed among phytopathogenic bacteria of the *Pectobacterium*, *Dickeya*, and *Xanthomonas* genera and are also present in a plant endophyte and several species of plant-non-associated bacteria (both free-living and associated with animals) from the *Gammaproteobacteria*, *Alphaproteobacteria*, and *Deltaproteobacteria* classes. According to the bioinformatic analysis, almost all (except one) of the revealed Svx-like proteins are either extracellular or plasma membrane-localized since they possess Sec/SPI or Tat/SPI or Sec/SPII signal peptides. For some of these proteins, their extracellular localization has been previously shown experimentally [12,14].

The Svx-like proteins of phytopathogenic species form a distinct clade on a phylogram. As such, Svx-like proteins from closer relatives with different eco-niches (*Pectobacterium*/*Dickeya* and *Vibrio*/*Teredinibacter*/*Microbulbifer*) appeared phylogenetically more distant than the Svx-like proteins from more distant relatives (*Pectobacterium*/*Dickeya* and *Xanthomonas*) with a common eco-niche—the host plant. The fact that the Svx-like proteins of *Gammaproteobacteria* split into two separate phylogenetic clades corresponding to the proteins of (1) phytopathogens (clade G in Figure 1) and (2) plant-non-associated bacteria (clade DG in Figure 1) points to the directed evolution of these proteins in plant pathogens for providing effective interactions with the host plants. Therefore, the Svx-like proteins from plant-non-associated bacteria can presumably be “designed” to implement functions other than those of the Svx-like proteins from phytopathogenic species. Thus, the Svx-like proteins are likely to be “multifaced” proteins that participate in distinct physiological/biochemical processes depending on the lifestyle of the bacteria producing them.

To get better insight into the structure and functions of the Svx-like proteins, we analyzed the Svx of *Pba*. The *Pba* Svx has been previously shown to be required for full virulence and to be transferred via T2SS, while the corresponding gene *svx* has been demonstrated to be upregulated by quorum sensing and plant metabolites [7,10,12,14]. In our study, in silico analysis predicted two domains of the *Pba* Svx protein: a peptidase domain and a transferase-like domain with an unknown function. The peptidase domain of the *Pba* Svx had a catalytic site formed by the conservative zinc-binding motif HEXXHX(8,28)E typical of the gluzincin extracellular metallopeptidases [32]. By obtaining and analyzing the recombinant *Pba* Svx protein and its two mutant forms with the amino acid substitutions (ΔE141A and ΔE167A) in the HEXXHX(8,28)E motif, we confirmed that the *Pba* Svx protein was a gluzincin metallopeptidase.

In addition, we found that *Pba* Svx has carbohydrate-binding residues typical of M60 family glycopeptidases (PF13402) that catalyze the hydrolysis of glycoproteins [39]. Therefore, we presumed that *Pba* Svx is a glycopeptidase that breaks down glycosylated PCW proteins since *Pba* Svx is secreted by bacteria into the host plant apoplast. To check our hypothesis, we built the spatial structure of the *Pba* Svx and performed molecular docking with the proposed substrates—the fragments of the glycosylated PCW proteins.

The tertiary structure of the peptidase domain of the *Pba* Svx was the most similar to that of the peptidase domain of the O-glycopeptidase ZmpB of *Clostridium perfringens*, which recognized N-acetyl-galactosamine residues attached to the polypeptide chain. Therefore, we performed the molecular docking of the *Pba* Svx protein with a ligand similar to that used for the study of ZmpB-substrate interaction, except that N-acetyl-galactosamine attached to threonine was substituted for the galactose attached to serine since there are no known PCW proteins in which N-acetyl-galactosamine directly bound to the polypeptide chain. In turn, α-galactose linked to serine of the polypeptide chain is a characteristic feature of some O-glycosylated PCW proteins, namely extensins or proteins with extensin domains.

The performed analysis showed that *Pba* Svx protein could accommodate an α-galactose glycosylated peptide typical of extensins with millimolar affinity, which is in agreement with the affinity of different lectins to the terminal α-galactose in glycolipids [55]. The binding of α-galactosylated peptide by *Pba* Svx occurred mostly due to the interactions with the sugar moiety.

Then, we investigated whether the PCW glycoproteins with a β-linkage between aglycon and glycoside could be the potential substrates of the *Pba* Svx. The structural-based analysis showed that both β-galactosylated hydroxyproline (a characteristic of O-glycosylated arabinogalactans) and asparagine β-linked to N-acetyl-glucosamine (a characteristic of N-glycosylated PCW proteins) brought the peptide away from Zn^2+^ and led to the overlapping of the ligand’s peptide part with the electron density of the *Pba* Svx. This means that the topology of the carbohydrate binding site of the *Pba* Svx protein hardly enables this enzyme to hydrolyze peptide linkage adjacent to the amino acid residue attached to the carbohydrate residue by the β-linkage. Thus, the extensins are the most probable substrates of the *Pba* Svx among the PCW glycosylated proteins.

Extensins are structural PCW proteins that interact with each other as well as with lignin and polysaccharides substituted with aromatic components, making the PCW more rigid [53]. Therefore, it is reasonable to conclude that the destruction of extensins promotes disease development caused by *Pba*. Moreover, some extensins (designated as LRX), containing an additional N-terminal domain composed of leucine-rich repeats (LRR), are considered important sensors that trace the state of PCW and transmit the signals via partner proteins [56,57]. Thus, breaking down or modifying extensins may not only weaken PCW but also influence the signaling processes of the host plant, leading to the repression of defense reactions or the induction of susceptible responses [58].

In plant pathogenic bacteria, metallopeptidases that are able to degrade extensins have been previously described: the Prt1 from *Pectobacterium carotovorum* and the gp120-degrading enzyme from *Xanthomonas campestris* [59,60]. Prt1 interacts with non-glycosylated ligands and catalyzes the hydrolysis of a wide range of proteins, such as casein, collagen, bovine serum albumin, ribonuclease A, potato lectin, and plant extensins [59]. In turn, gp120-degrading metallopeptidase from *X. campestris* shows specific enzymatic activity toward particular glycosylated proteins, such as potato and tomato extensins and glycoproteins gpS-3 and gp120, but does not catalyze the hydrolysis of casein or gp45 [60]. Our findings show that the *Pba* Svx protein’s carbohydrate binding site is organized in such a way that it can interact with α-glycosylated peptides (including extensins); thus, *Pba* Svx can also catalyze the hydrolysis of non-glycosylated substrates like azocasein.

When taken together, our results show that *Pba* Svx is a gluzincin metallopeptidase. Furthermore, in silico studies indicate that its substrate-binding pocket may accept α-glycosylated protein substrates represented in the PCW by extensins and proteins containing an extensin domain. Our future work will be focused on the physiological action of the *Pba* Svx protein, characterization of the Svx proteins from other species, and getting better insights into the functions of the second revealed functional domain—the acyltransferase-like one. It should be noted that this domain lacks the amino acid residues required for acyltransferase activity, so it is unlikely to have catalytic activity. It is possible that the acyltransferase-like domain of the *Pba* Svx provides the formation of the protein multimers or takes part in substrate recognition.

## 4. Materials and Methods

### 4.1. Phylogenetic Analysis

The search of amino acid sequences of the Svx-homologs was performed in the RefSEq Select (Reference protein) NCBI database [61] with the DELTA-BLAST (Domain Enhanced Lookup Time Accelerated BLAST) algorithm [15,16]; the amino acid sequence of the *Pectobacterium atrosepticum* SCRI1043 Svx protein (WP_011092533.1) was used as a reference sequence. The amino acid sequences of the Svx-like proteins were aligned using the MAFFT algorithm [62,63] and the BLOSUM62 substitution matrix. A phylogenetic tree was constructed using the Poisson model and the minimal evolution algorithm in MegaX [64,65]). The statistical support of the phylogenetic tree was estimated by bootstrap analysis [26], with 1000 replicates. Identification of signal peptides in Svx-homologs was performed using the SignalP 6.0 server [66,67].

### 4.2. Prediction of Functional Domains

The prediction of functional domains of the Svx-homolog proteins was carried out using servers NCBI Conserved Domain Search [68,69], HMMER [70,71], Phyre2 [30,31], and I-Tasser [28,29].

### 4.3. Gene Cloning and Protein Purification

The *svx* gene (ECA0931) without a signal-peptide coding sequence was PCR-amplified from the genomic DNA of *Pectobacterium atrosepticum* SCRI1043 using the primers SvxNco_R: CAGCTACCATGGCCGCTGAAGCTTGCG and SvxSac_F: GTGCTAGAGCTCTCATCACTTTTCGAACTGCGGGTGGCTCCATTTCGTGCTGTAGACC. The obtained PCR product with C-terminal Strep-tag II was ligated into the pET-51b vector. Cells of *Escherichia coli* strain BL21 (DE3) (Invitrogen, Waltham, MA, USA) were transformed with the resultant pET51b-Svx plasmid and grown at 37 °C in 2 L of sterile LB:M9 (1:1) media supplemented with 100 µg/mL ampicillin until the culture reached a density of ~0.6 at 600 nm. Recombinant protein synthesis was induced by the addition of isopropyl-β-D-1-thiogalactopyranoside (IPTG) (Thermo Fisher Scientific, Waltham, MA, USA) to a final concentration of 0.5 mM. Cultures were kept overnight at 18 °C with shaking. Bacterial cells were collected, and cell lysis was performed using BugBuster Protein Extraction Reagent (Novagen, Temecula, CA, USA), 0.1 mg/mL of lysozyme (Thermo Fisher Scientific, Waltham, MA, USA), and 0.1 mg/mL of DNaseI (Bio-Rad, Hercules, CA, USA) at 37 °C for 20 min. Cell lysates were centrifuged (12,000× *g*, 4 °C, 30 min), and the insoluble fraction (inclusion bodies) was washed with the buffer (100 mM Tris-HCl, pH 8.0, 150 mM NaCl) and pelleted again (12,000× *g*, 4 °C, 30 min). The pellet was dissolved in a denaturing buffer (100 mM Tris-HCl, 150 mM NaCl, 8 M urea). Then, one volume of resuspended inclusion bodies was added dropwise to 20 volumes of the refolding buffer (100 mM Tris-HCl, 150 mM NaCl, 1.25 mM reduced glutathione, 1.25 mM oxidized glutathione, 0.25 M arginine, and 1 mM ZnSO_4_) at 4 °C for 12 h. The resultant solution was clarified by centrifugation at 10,000× *g* for 30 min. The refolded protein in parts was loaded on the chromatography column with Strep-Tactin Superflow High Capacity resin (IBA-Lifescienses, Göttingen, Germany). The one-step purification was carried out at 4 °C. The *Pba* Svx protein was eluted by the buffer (100 mM Tris-HCl, 150 mM NaCl, 10 mM desthiobiotin) and used for enzymatic assays.

### 4.4. Site-Directed Mutagenesis

Substitution of amino acids in the active site of the *Pba* Svx protein (yielding mutant proteins SvxΔE141A and SvxΔE167A) was performed using the site-directed mutagenesis. The site-directed mutagenesis was performed as described in QuikChange II Site-Directed Mutagenesis Kit Instruction manual (Agilent Technologies, Santa Clara, CA, USA) [72]) using Q5 High Fidelity DNA polymerase (NEB, Ipswich, MA, USA) for primer-directed replication of both plasmid strands.

### 4.5. Peptidase Activity Assays

Peptidase activity assays were carried out using azocasein as a substrate (Sigma-Aldrich, St. Louis, MO, USA) according to the previously described method [73]. Ten μg (in 100 μL) of the recombinant *Pba* Svx protein (or its mutant forms SvxΔE141A and SvxΔE167A) was incubated for 60 min in 250 μL of 0.5% azocasein solution in 100 mM Tris-HCl buffer with a pH range from 7 to 9 and a temperature range from 20 °C to 50 °C. When specified, the reaction mixtures were supplemented with 1 mM or 2 mm ZnSO_4_ and/or 1 mM EDTA. The reactions were stopped by the addition of 100 μL of 10% trichloroacetic acid (Sigma-Aldrich, St. Louis, MO, USA). The sediment was removed by centrifugation (5000× *g*, 10 min, 25 °C). Four hundred μL of the supernatant and 133 μL of 1.0 N NaOH were mixed, then the absorbance at 440 nm was measured using the microplate reader CLARIOstar (BMG Labtech GmbH, Ortenberg, Germany). One unit of peptidase activity was defined as the amount of enzyme required to produce an absorbance change of 0.1 per min per 1 mg of the *Pba* Svx protein.

### 4.6. Circular Dichroism Spectroscopy

CD spectroscopy was used to analyze the secondary structure of the *Pba* Svx protein. Far-UV CD measurements were carried out using a Jasco J-1500 spectro-polarimeter (Jasco Corporation, Tokyo, Japan) equipped with a Peltier temperature controller unit. The spectra were registered every 1 nm in the spectral range of 190–250 nm, in quartz cuvettes with a path length of 0.1 cm, with an acquisition time of 1 s per point, at 22 °C. Each spectrum was obtained from an average of 3 scans. The protein concentrations were 0.1, 0.2, and 0.4 mg/mL for far-UV CD. The baseline was corrected in all experiments by using a protein-free 10 mM Tris-HCl, pH 8.0 solution as a control.

The ellipticity reading for each wavelength was presented as mean molar ellipticity by residue (MRE) according to the following equation: MRE = 100 × θ × M/C × n × L. In this equation, θ is the ellipticity in degrees; n, the number of amino acid residues; C, protein concentration (mg/mL); M, molecular mass; and L, the optical path (cm). The value of MRE was expressed in deg·cm^2^dmol^−1^. The content of the secondary structures in the protein was estimated by the analysis of the deconvolution of the far-UV CD spectra according to the CONTINLL algorithm [47,48]. Deconvolution was carried out using an online DICHROWEB Web server [74,75], and the reference protein database 7 was set [76].

### 4.7. In Silico Structure Prediction and Protein-Ligand Docking

The prediction of the *Pba* Svx protein structure was carried out using the AlphaFold 2 service [45,46]). Structure verification was performed based on the data from the PROCHECK server [49,50]. The PSIPRED server [43,44] was used to predict the secondary structure based on the primary structure. Protein-ligand docking was performed with the AutoDock4.2 server [77,78]. The ligand for the *Pba* Svx protein was built based on the structure of the ligand previously used for the study of the interaction of ZmpB glycopeptidase from *Clostridium perfrigens* with the substrate (pdb code 5KDS) [39]; thus, N-acetyl-galactosamine moiety linked to threonine was replaced by galactose linked to serine. The affinity of the *Pba* Svx protein to the ligand was calculated based on the binding energy estimated by docking. In AutoDock4.2, the binding energy was a sum of electrostatic interactions between the protein and the ligand, hydrogen bonds, van der Waals contacts, desolvation energy, and the contribution from ligand conformational entropy.

## Figures and Tables

**Figure 1 ijms-23-06914-f001:**
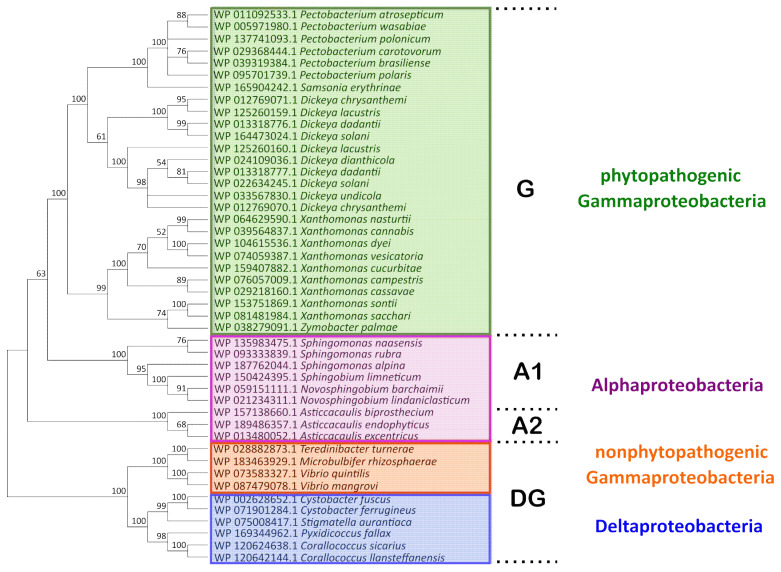
The phylogenetic tree of the amino acid sequences of Svx-like proteins. The sequences were obtained from the NCBI RefSeq Select Protein Database and aligned using the MAFFT algorithm and the BLOSUM62 substitution matrix. The cladogram was built using the Poisson model and the minimal evolution algorithm in MegaX. The statistical support of the phylogenetic tree was estimated by bootstrap analysis [26], with 1000 replicates. G, A1, A2, and DG designate separate clades described in the text.

**Figure 2 ijms-23-06914-f002:**
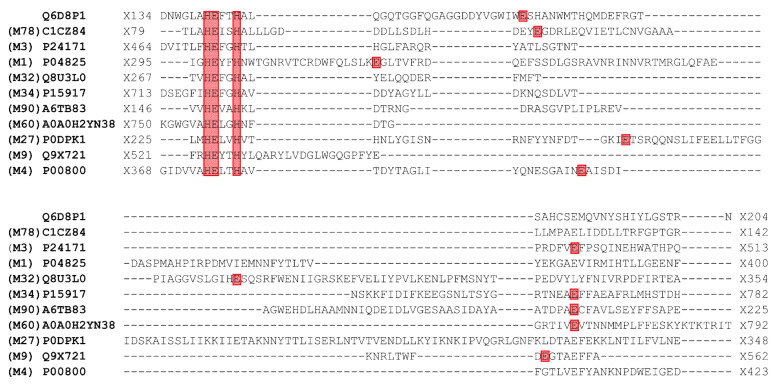
The alignment of the amino acid sequence of the Svx protein of *Pectobacterium atrosepticum* (*Pba*) (UniProt ID: Q6D8P1) with other bacterial gluzincin metallopeptidases with experimentally determined spatial structures: radiation response metallopeptidase IrrE *Deinococcus deserti* (UniProt ID: CICZ84) MEROPS family M78 [33], dipeptidyl carboxypeptidase *Escherichia coli* (UniProt ID: P24171) MEROPS family M3 [34], aminopeptidase N *Escherichia coli* (UniProt ID: P04825) MEROPS family M1 [35], thermostable carboxypeptidase 1 *Pyrococcus furiosus* (UniProt ID: Q8U3L0) MEROPS family M32 [36], lethal factor *Bacillus anthracis* (UniProt ID: P15917) MEROPS family M34 [37], protein MtfA *Klebsiella pneumoniae* (UniProt ID: A6TB8) MEROPS family M90 [38], ZmpB O-glycopeptidase *Clostridium perfringens* (UniProt ID: A0A0H2YN38) MEROPS family M60 [39], botulinum neurotoxin type X *Clostridium botulinum* (UniProt ID: P0DPK1) MEROPS family M27 [40], collagenase ColG *Hathewaya histolytica* (UniProt ID: Q9X721) MEROPS family M9 [41], thermolysin *Bacillus thermoproteolyticus* (UniProt ID: P00800) MEROPS family M4 [42]. The conservative zinc-binding motif HEXXHX(8,28)E is colored in red (the positions of the last glutamic acid (E) residues are variable in different proteins). The alignment was built using the MAFFT algorithm and the BLOSUM45 substitution matrix.

**Figure 3 ijms-23-06914-f003:**
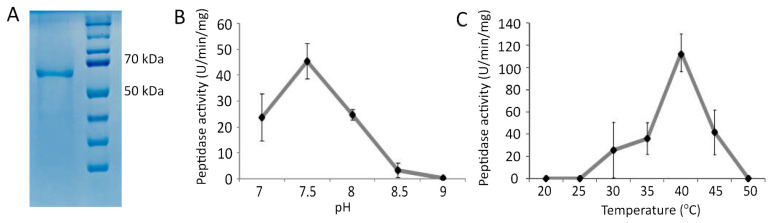
The purified recombinant Svx protein of *Pectobacterium atrosepticum* (66 kDa) (**A**) and its peptidase activity at different pH values (**B**) and different temperatures (**C**). The pH optimum for the enzymatic activity was determined in 100 mM Tris-HCl buffer at 37 °C. The temperature optimum was determined in 100 mM Tris-HCl buffer pH 7.5.

**Figure 4 ijms-23-06914-f004:**
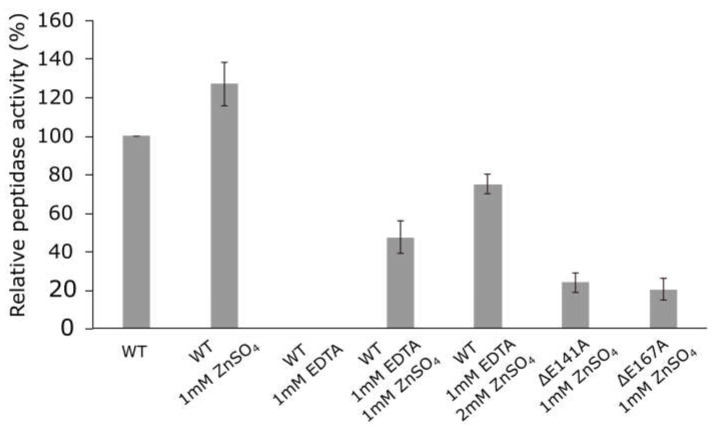
Peptidase activity of the wild-type Svx protein of *Pectobacterium atrosepticum* and its mutant forms SvxΔE141A and SvxΔE167A. The activities of the wild-type protein were measured in the absence (WT) or presence of 1 mM ZnSO_4_ (WT 1 mM ZnSO_4_) or 1 mM EDTA (WT 1 mM EDTA) or both 1 mM EDTA and either 1 mM or 2 mM ZnSO_4_ (WT 1 mM EDTA (1 or 2) mM ZnSO_4_). The activities of the mutant proteins were measured in the presence of 1 mM ZnSO_4_ (ΔE141A 1 mM ZnSO_4_/ΔE167A 1 mM ZnSO_4_). All reactions were carried out in 100 mM Tris-HCl buffer pH 7.5 at 40 °C.

**Figure 5 ijms-23-06914-f005:**
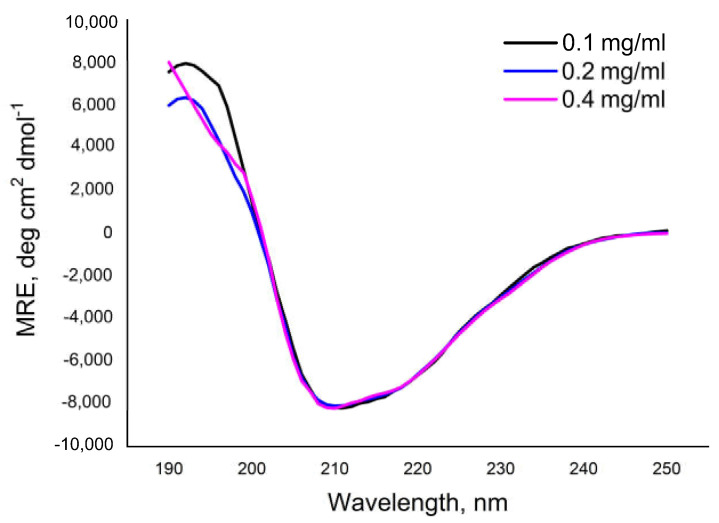
The circular dichroism spectra of the Svx protein of *Pectobacterium atrosepticum* in 10 mM Tris-HCl buffer (pH 8.0).

**Figure 6 ijms-23-06914-f006:**
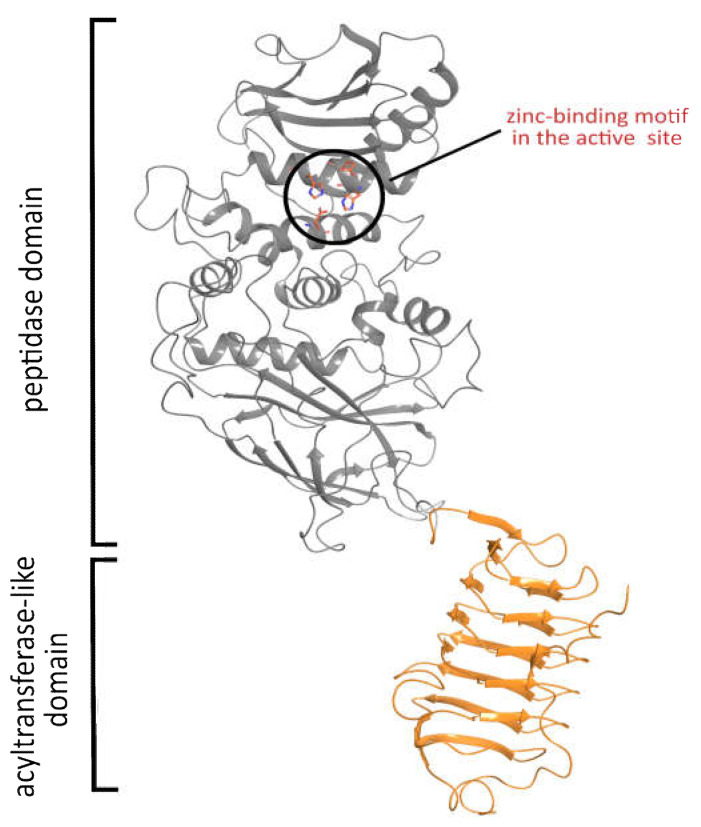
The spatial structure of the Svx protein of *Pectobacterium atrosepticum* predicted by AlphaFold 2. The built model demonstrated the presence of two functional domains: peptidase and acyltransferase-like.

**Figure 7 ijms-23-06914-f007:**
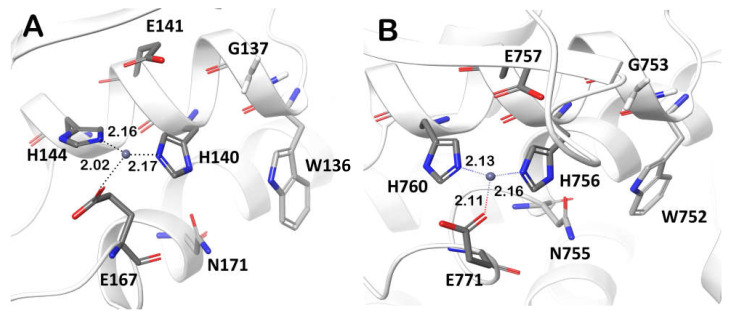
The conserved residues forming the zinc-binding motif HEXXHX(8,28)E and the carbohydrate binding site (W, G, N) in the active sites of the peptidase domains of the Svx protein of *Pectobacterium atrosepticum* (**A**) and the O-glycopeptidase ZmpB of *Clostridium perfringens* (pdb code: 5KDS) (**B**).

**Figure 8 ijms-23-06914-f008:**
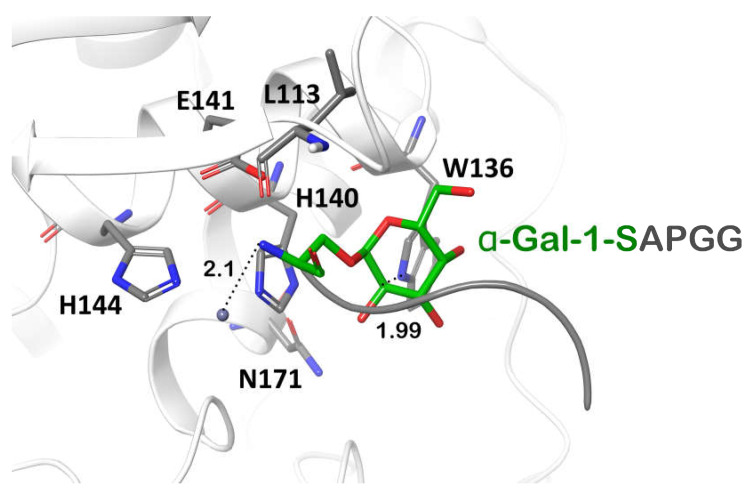
The structure of the peptidase domain active site of the Svx protein of *Pectobacterium atrosepticum* with a α-Gal-1-SAPGG ligand. The ligand was built based on the structure of the ligand previously used for the study of ZmpB-substrate interaction (pdb code 5KDS) [39].

**Figure 9 ijms-23-06914-f009:**
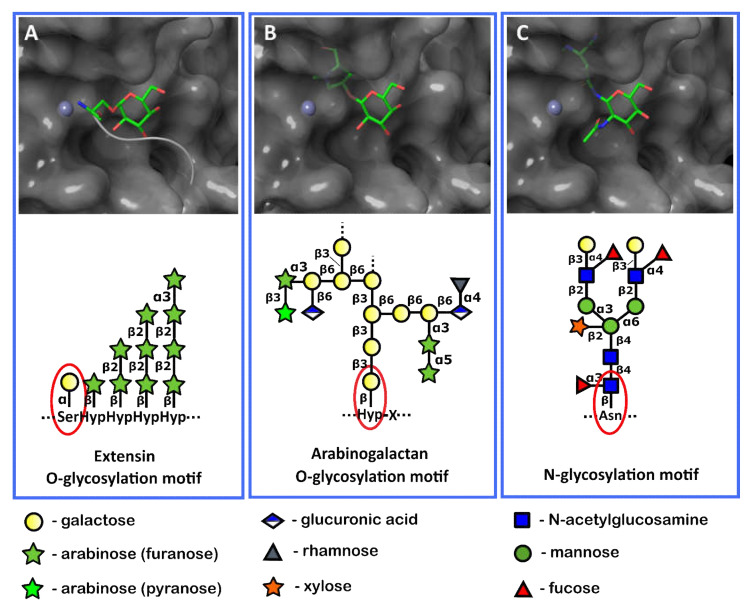
The orientation of the α-glycosylated (**A**) and β-glycosylated (**B**,**C**) amino acid residues characteristic of the extensin (**A**) or arabinogalactan (**B**) O-glycosylation motifs and plant protein N-glycosylation motif (**C**) in the pocket of the active site of the Svx protein of *Pectobacterium atrosepticum*. Red ellipses show the sugar residues attached to the amino acid residues in the plant cell wall glycoproteins. These parts of glycoproteins were used as ligands for docking and superposition with the *Pba* Svx protein.

**Table 1 ijms-23-06914-t001:** The percentage of the secondary structure elements of the Svx protein of *Pectobacterium atrosepticum* calculated based on the circular dichroism spectroscopy data, AlphaFold 2 modeling and PSIPRED.

	CD, %	AlphaFold 2, %	PSIPRED, %
Helix (α-helices)	17.0 ± 0.49	19.1	18.0
Strand (β-sheets)	32.3 ± 0.98	24.0	21.5
Turns	21.2 ± 0.28	30.0	60.5
Unordered	29.5 ± 0.86	26.9	0

## Data Availability

Not applicable.

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
