# Peer review of "Structure-Functional Characteristics of the Svx Protein—The Virulence Factor of the Phytopathogenic Bacterium Pectobacterium atrosepticum"

_ijms, 2022, doi:10.3390/ijms23136914_

Round 1

Reviewer 1 Report

Major comment

In this study, Svx proteins which is involved in plant pathogenesis with unknown mechanism, was studied to predict the protein function mainly in silico. The target protein was extesively studied to predict the molecular function. The manuscript was written clearly. They demonstrate that the protein has protease activity by using azocasein as the substrate, and hypothesized that it is a glycopeptidase through finding of carbohydrate binding domain and similarity between the model structure and ZmpB. However, molecular docking simulations were performed to evaluate the hypothesis. It means in silico prediction was supported by in silico prediction. I believe it's not good enough to conclude that Pba Svx is "Pba Svx is a gluzincin metallopeptidase with a substrate-binding pocket that accepts α-glycosylated protein substrates (page 12)". Therefore I would like to request to perform the enzyme assay with substrates which were used for ZmpB (ref 39) or extensins. 

Minor comment

1) I sometimes confuse what is "the target protein". Please check it out.

2) In the peptidase activity assay in Fig. 4, only EDTA is necessary to conclude that the enzyme require metal ion. 

3) In page 7, the second domain predicted as a transferase was suddenly came out. Please explain how it was found. 

4) Three amino acids W136, G137 and N171 are predicted for carbonhydrate binding site in page 8. Please indicate them in Fig. 2.

5) What is protease family for Pba Svx?

Author Response

Thank you for your suggestions to improve the manuscript. Please, find below point-by-point answers to your comments.

Major comment

In this study, Svx proteins which is involved in plant pathogenesis with unknown mechanism, was studied to predict the protein function mainly in silico. The target protein was extesively studied to predict the molecular function. The manuscript was written clearly. They demonstrate that the protein has protease activity by using azocasein as the substrate, and hypothesized that it is a glycopeptidase through finding of carbohydrate binding domain and similarity between the model structure and ZmpB. However, molecular docking simulations were performed to evaluate the hypothesis. It means in silico prediction was supported by in silico prediction. I believe it's not good enough to conclude that Pba Svx is "Pba Svx is a gluzincin metallopeptidase with a substrate-binding pocket that accepts α-glycosylated protein substrates (page 12)". Therefore I would like to request to perform the enzyme assay with substrates which were used for ZmpB (ref 39) or extensins.

Indeed, only the fact that Pba Svx is a gluzincin metallopeptidase was proved using the in vitro studies (the recombinant enzyme). The possibility of accepting α-glycosylated protein substrates by the Pba Svx was shown by in silico studies. However, to prove this using in vitro studies, the representative extensins (which are represented by large glycoprotein families) should be chosen, and the corresponding genes should be cloned to obtain the recombinant extensins that would be used as substrates for the Pba Svx. We consider it a further separate study. The use of the ZmpB substrate (mucin) is unlikely to be a good option since the mucin is of animal origin and is not a "universal" substrate accepted by all glycopetidases. We agree with the reviewer that in the initial version of the manuscript, the conclusion looked over-interpreted; therefore, we have rephrased the conclusion to make it suitable for the results obtained.

Minor comment

  1. I sometimes confuse what is "the target protein". Please check it out.

The target protein means the protein of interest – the Svx protein of Pectobacterium atrosepticum. In the revised version we replaced "the target protein" with the Pba Svx protein.

  1. In the peptidase activity assay in Fig. 4, only EDTA is necessary to conclude that the enzyme require metal ion.

EDTA is a popular chelating agent for divalent metal ions. The decrease of peptidase activity in the presence of this inhibitor supports the idea that the Pba Svx protein requires divalent metal ions for its enzymatic activity. The low peptidase activity of mutant forms of the Pba Svx with amino acid substitutions in the zinc-binding motif also demonstrates the importance of the presence of zinc ions, as the binding of divalent zinc to protein is hampered in the absence of appropriate amino acid radicals.

  1. In page 7, the second domain predicted as a transferase was suddenly came out. Please explain how it was found.

The acyltransferase-like domain was found by analyzing the amino acid sequences of the Svx protein and its homologues in the NCBI Conserved Domain Search, HMMER, and Phyre2 servers (described in the text). The model of the tertiary structure of Svx built by AlphaFold 2 supported the idea of the presence of a second domain that has a fold similar to acyltransferases of the LpxA or LpxD families. In the revised version, we added the supplementary figure S1 showing the functional domains predicted for Svx-homologues by NCBI Conserved Domain Search, HMMER and Phyre2 servers.

  1. Three amino acids W136, G137 and N171 are predicted for carbonhydrate-binding site in page 8. Please indicate them in Fig. 2.

The main idea of Figure 2 was to indicate the amino acid residues of the zinc-binding motif, which the Pba Svx protein shares with other gluzincin metallopeptidases. To show the similarity of carbohydrate–binding sites of the Pba Svx and M60 metallopeptidases in the revised version, we also provided another alignment in supplementary Figure S3.

5) What is protease family for Pba Svx?

The assignment of the Pba Svx to a particular protease family requires additional experiments on the mechanism of its catalytic action. We can only speculate that the Pba Svx may belong to the O-glycopeptidases from the M60 family based on the similarity of the amino acid sequence of the peptidase domain and the structure of its active site in the Pba Svx and O-glycopeptidases from M60 (the alignment is provided in the supplementary Figure S3).

Reviewer 2 Report

An interesting study was conducted in this manuscript aiming to predict the function and structure of the protein Svx from a phytopathogenic bacterium. Methodology includes enzymatic assays, DC and phylogenetic studies as well as predictions of structure and ligand interaction. It is an interesting approach that sheds light on the likely function of this virulence factor.  The assays are thoroughly described, and the predictions seem reasonable.  This well-written manuscript makes it easy to follow the research without additional effort.

Having said that, I believe that some remarks could be useful in improving this work:

- The optimum temperature was determined to be 40 °C. However, neither the results nor the discussion point out the reason why that peptidase activity is enhanced at that temperature.  Are there other temperature-based studies with similar conclusions? Can the author elaborate on this?  Clearly, the biological conditions in host plants are different from those in vitro, and the optimal T could vary.  Likewise, in this point, determining the Tm of this enzyme would be interesting (performing for instance an ITC experiment…)

-           In this study, two mutants are generated on glutamic residues of the conserved zinc-binding motif.  Would a double mutant be feasible?  Does the author consider that an enzyme with the two glutamic residues replaced would lose its activity?

- I miss the confidence parameters that come from Alpha Fold 2 after the prediction that support the model's reliability. The relative spatial organisation between both domains displayed on figure 6 is the one predicted or could be another?

- In the Figure 9 legend, it is advisable to show what the red ellipses mean.

Author Response

Thank you for your suggestions to improve the manuscript. Please, find below point-by-point answers to your comments.

  1. The optimum temperature was determined to be 40 °C. However, neither the results nor the discussion point out the reason why that peptidase activity is enhanced at that temperature. Are there other temperature-based studies with similar conclusions? Can the author elaborate on this? Clearly, the biological conditions in host plants are different from those in vitro, and the optimal T could vary. Likewise, in this point, determining the Tm of this enzyme would be interesting (performing for instance an ITC experiment…).

Such an effect is rather common for the activity of recombinant enzymes. For example, the temperature optimum for the metallopeptidase Prt1 of Pectobacterium carotovorum activity in vitro was 50 °C (doi.org/10.1007/s00253-014-5877-2), while for the phenoloxidase of Pectobacterium atrosepticum it was 60 °C (doi.org/10.1002/jobm.201700413). Such a common effect can be presumably explained by the influence of temperature on the thermal motion of molecules that makes enzyme-substrate complexes more effective in vitro. Furthermore, in vivo, the increased enzymatic activity can be provided by unknown cofactors whose actions cannot be replicated in vitro.

  1. In this study, two mutants are generated on glutamic residues of the conserved zinc-binding motif. Would a double mutant be feasible? Does the author consider that an enzyme with the two glutamic residues replaced would lose its activity?

The double mutant will lose its activity with a high probability because one of the glutamic acids is required for zinc binding and another for nucleophile water molecule coordination, which is obligatory for the catalysis.

  1. I miss the confidence parameters that come from Alpha Fold 2 after the prediction that support the model's reliability. The relative spatial organisation between both domains displayed on figure 6 is the one predicted or could be another?

The information about the predicted local-distance difference test (pLDDT) is added as the supplementary figure S2 and described in the revised version of the manuscript. The model confidence is high, with LDDT above 70% for the majority of the residues. The region of low confidence (slightly below 70%) is located on the loop.

We also provide the coordinates of the structure built up by Alpha Fold 2 in the supplementary file. In the original structure, the acyltransferase-like domain forms contact with the metallopeptidase domain. The possible role of the acyltransferase-like domain in conjunction with the metallopeptidase domain still remains to be established. For the clarity of illustration, since in this paper we focused on the metallopeptidase domain, the mutual orientation of the domains was modified (by displacement of the acyltransferase-like domain). Since the linker that connects two domains (about 10 aa) is flexible and no specific interdomain interactions are visible, we assume that a variety of spatial arrangements between two domains are possible.

  1. In the Figure 9 legend, it is advisable to show what the red ellipses mean.

In Figure 9, red ellipses show the sugar residues attached to the amino acid residues in the plant cell wall glycoproteins. These parts of glycoproteins were used as ligands for docking and superposition with the Pba Svx protein. The clarification is now given in the figure legend.

Reviewer 3 Report

The paper presents a detailed examination of the structure and functions of the Svx proteins. In order to improve the paper just I have a few suggestions:

 Besides using PROCHECK to check the Alphafold2 model it would be useful to know the Alphafold2 reliability score of the model of the Svx protein of Pectobacterium atrosepticum?

Given that Alphafold2 is an AI-based algorithm, it would be interesting to know how well Alphafold2 performs on other metallpopeptidases?

Also, could the authors make available the Alphafold2 model(s) generated?

Author Response

Thank you for your suggestions to improve the manuscript. Please, find below point-by-point answers to your comments.

  1. Besides using PROCHECK to check the Alphafold2 model it would be useful to know the Alphafold2 reliability score of the model of the Svx protein of Pectobacterium atrosepticum?

The information about the predicted local-distance difference test (pLDDT) is included (supplementary Figure S2) and described in the revised version of the manuscript. The model confidence is high, with LDDT above 70% for the majority of the residues. The region of low confidence (slightly below 70%) is located on the loop.

  1. Given that Alphafold2 is an AI-based algorithm, it would be interesting to know how well Alphafold2 performs on other metallopeptidases?

The coverage of sequence similarity (multiple sequence alignment returned by Alpha Fold) is around 500 for almost full sequence, the aa region 100 - 500. For example, the AlphaFold predicted structure of thermolysin (zinc metallopeptidase from M4 family) shares high similarity value with its experimentally obtained structure with resolution 1,12 A (RMSD 0.282).

  1. Also, could the authors make available the Alphafold2 model(s) generated?

We added the coordinates of the structure built up by Alpha Fold 2 (supplementary file 1).

Round 2

Reviewer 1 Report

The revised manuscript was edited following the comments from reviewers, and I accept the revises except EDTA. Yes, EDTA is a popular chelating agent for divalent metal ions. One molecule of EDTA binds to one metal atom. Therefore, chelating titer of 1mM EDTA is ideally extinguished by 1mM ZnSO4. Probably, the lower enzyme activity is caused by pH or ion strength changing or else. So, I request to do the enzyme assay with EDTA only. For instance, if 1mM EDTA extinguished the enzyme activity and 1mM EDTA + 2mM ZnSO4 restored the activity, the result proved that Zn2+ was necessary for the enzyme activity. 

Author Response

The revised manuscript was edited following the comments from reviewers, and I accept the revises except EDTA. Yes, EDTA is a popular chelating agent for divalent metal ions. One molecule of EDTA binds to one metal atom. Therefore, chelating titer of 1mM EDTA is ideally extinguished by 1mM ZnSO4. Probably, the lower enzyme activity is caused by pH or ion strength changing or else. So, I request to do the enzyme assay with EDTA only. For instance, if 1mM EDTA extinguished the enzyme activity and 1mM EDTA + 2mM ZnSO4 restored the activity, the result proved that Zn2+ was necessary for the enzyme activity.

Thanks for your suggestion. We performed the additional experiment with EDTA only as well as in the presence of both 1 mM EDTA and 2 mM ZnSO4. EDTA totally inhibited the peptidase activity of the Pba Svx protein, while 1 mM or 2 mM ZnSO4 gradually restored the peptidase activity of Pba Svx protein in the presence of 1 mM EDTA. The results are included in the revised version of the manuscript.